# Vestibular Disorders and Hormonal Dysregulations: State of the Art and Clinical Perspectives

**DOI:** 10.3390/cells12040656

**Published:** 2023-02-18

**Authors:** Rhizlane El Khiati, Brahim Tighilet, Stéphane Besnard, Christian Chabbert

**Affiliations:** 1Aix Marseille University—Centre National de la Recherche Scientifique (CNRS), Laboratory of Cognitive Neurosciences, UMR7291, Team Pathophysiology and Therapy of Vestibular Disorders, 13331 Marseille, France; 2Research Group on Vestibular Pathophysiology, Centre National de la Recherche Scientifique (CNRS), Unit GDR2074, 13331 Marseille, France

**Keywords:** vestibular disorders, sex hormones, thyroid, diabetes

## Abstract

The interaction between endocrine and vestibular systems remains poorly documented so far, despite numerous observations in humans and animals revealing direct links between the two systems. For example, dizziness or vestibular instabilities often accompany the menstrual cycle and are highly associated with the pre-menopause period, while sex hormones, together with their specific receptors, are expressed at key places of the vestibular sensory network. Similarly, other hormones may be associated with vestibular disorders either as causal/inductive factors or as correlates of the pathology. This review was carried out according to the PRISMA method, covering the last two decades and using the MEDLINE and COCHRANE databases in order to identify studies associating the terms vestibular system and/or vestibular pathologies and hormones. Our literature search identified 646 articles, 67 of which referred directly to vestibular dysfunction associated with hormonal variations. While we noted specific hormonal profiles depending on the pathology considered, very few clinical studies attempted to establish a direct link between the expression of the vestibular syndrome and the level of circulating hormones. This review also proposes different approaches to shed new light on the link between hormones and vestibular disorders, and to improve both the diagnosis and the therapeutic management of dizzy patients.

## 1. Introduction

The interaction between endocrine and vestibular systems remains poorly documented so far, despite numerous observations in humans and animals revealing direct links between both systems.

For example, it has long been known that receptors of different types of hormones, such as adrenaline [1], vasopressin [1,2], thyroid hormones [3], insulin [4], cortisol [5] or sex hormones, such as testosterone, progesterone, and estrogens [6,7,8], are expressed at the level of the inner ear sensors, along the vestibular nerve, and in the brainstem vestibular nuclei [5]. Some hormones, such as progesterone, estradiol, testosterone, and aldosterone have been found in the epithelial cells of the endolymphatic sac in mammals [8,9,10]. Since the development of a rodent model of hydrops [2,11,12,13,14], it has been believed that vasopressin may also actively participate in Ménière’s disease pathogenesis by altering the endolymph water flow in the inner ear, through involving vasopressin receptor 2 (V2R) and aquaporin 2 (AQP2) [11]. Moreover, vasopressin was suggested as a factor of motion sickness [15,16,17]. Several observations reveal that vestibular stimulation activates secretory central nuclei, that in turn trigger endocrine variations [18,19]. Thus, stimulation of the vestibular pathways induces paraventricular nucleus (PVN) [15,18] and supra-optic nucleus [20] responses. The latter secrete vasopressin, involved in Ménière’s disease pathogenesis [2]. The thyrotropin releasing hormone (TRH) neurons of the paraventricular hypothalamic nuclei (PVN) also play an essential role in the function of the hypothalamic–pituitary–thyroid axis, since they stimulate the secretion of TSH [21]. We know that steroids are directly involved in both the facilitating and deleterious effects of stress on vestibular compensation [22]. Moreover, a link between the vestibular system and chronobiological functions has been highlighted [23,24,25,26,27]. The vestibular system displays projections towards the suprachiasmatic (SCN) and raphe nuclei [28] involved in the secretion of serotonin [29] and melatonin [30], as well as bidirectional projections with orexinergic neurons [31,32]. In turn, the vestibular organs receive inputs from melatonin neurons [33]. Whether the conditions for a modulating action of these hormones on the detection, encoding, and integration of vestibular sensory information are therefore met, the molecular mechanisms supporting these actions, as well as their consequences on the vestibular function remain in most cases to be determined.

It has also been known for a long time that certain physiological states displaying particular hormonal profiles are commonly associated with vestibular disorders. This is the case, for example, with the strong preponderance of benign paroxysmal positional vertigo (BPPV) in postmenopausal women [34,35,36,37,38,39,40,41,42,43,44]. This is also the case for thyroid pathologies [45,46,47,48], diabetes [49,50,51,52,53], and vestibular migraines [54,55]. We also know that vestibular stimulation activates the sympathetic system [56] and the hypothalamic–pituitary axis with modulation of the secretions of CRH, ACTH, LHRH, and LH [57]. Additionally, metabolic syndromes such as hypo/hyperthyroidism and type I diabetes may affect vestibular organ and functions [58,59].

While we can multiply the examples of comorbidity between specific hormonal profiles and vestibular disorders, the precise role of circulating hormones as triggers, accompanying factors, or reactive mechanisms to vestibular disorders remains to be determined. Together, these structural, functional, and epidemiological observations suggest a more or less direct link between hormones and vestibule. Yet very few studies have attempted to detail the mechanisms of these interactions.

This study was designed to bring together the available information on two main questions: do patients with peripheral vestibular disorders have a specific hormonal profile compared to the healthy population? Do these patients display changes in circulating hormone levels during the acute episodes?

## 2. Methods

We carried out a bibliographic study according to the PRISMA method covering the last 20 years, from January 2001 to December 2021, on the PubMed and Cochrane platforms in order to identify studies associating the following keywords: vestibular system and hypothalamus, vestibular system and growth hormone, vestibular dysfunction and estrogen, vestibular dysfunction and menopause, vestibular dysfunction and cortisol, vestibular dysfunction and vasopressin, vestibular dysfunction and insulin/acute vertigo and hormones, acute vertigo and cortisol, acute and thyroid vertigo, acute vertigo and insulin, acute vertigo and progesterone, acute vertigo and estrogen, acute vertigo and vasopressin/BPPV and estrogen, BPPV and peri-menopause, BPPV and postmenopausal, BPPV and thyroid, BPPV and cortisol, BPPV and stress hormones, BPPV and vasopressin/vestibular migraine and hormones, vestibular migraine and menopause, vestibular migraine and estrogen, vestibular migraine and menstrual cycle, vestibular migraine and thyroid, vestibular migraine and insulin, vestibular migraine and cortisol/Ménière disease and stress hormones, Ménière disease and cortisol, Ménière disease and vasopressin, Ménière disease and estrogen, Ménière disease and menopause, Ménière disease and prolactin, Ménière and aldosterone. In total, 646 articles were collected. After excluding publications that did not refer directly to hormonal variations, 77 were retained for analysis. After removing duplicates, 67 articles were retained for our review. The steps of the data collection process according to the PRISMA method are described in Figure 1. The research method and the references of the studies collected are presented in Table 1.

## 3. Results

We identified a total of 67 articles on the interaction between vestibular dysfunction and hormonal changes. The time course of publications on the subject is presented in Figure 2. We note that the volume of publications differs according to the different pathologies and that a hormonal specificity exists according to the pathology studied. The results are shown in Table 1.

Fourteen papers were devoted to patients with BPPV, among which 11 investigated the impact of menopause and sexual hormonal changes in these patients [34,35,36,37,38,39,40,41,42,43,44]. Two papers addressed the combination of thyroid disease and BPPV [47,48], and one article investigated BPPV promoted by hyperglycemia [51]. Additionally, 24 of the 64 articles explored hormonal changes in patients with Ménière disease. Among them, 12 studied the variations of vasopressin [2,56,59,60,61,62,63,64,65,66,67,68], 2 those of prolactin [69,70], 1 melatonin [71], 1 aldosterone [72], 4 those of sex hormones [73,74,75,76], and 4 of stress hormones [77,78,79,80]. We found only two works on the correlation between vestibular migraine and estrogen/menopause [54,55]. No other hormone has been explored.

A more general search encompassing vestibular dysfunctions without specifying the pathology found 11 articles including five on vasopressin [2,12,13,15,81], two on insulin [52,53], seven on sex hormones [82,83,84,85,86,87,88], and one on cortisolemia and its impact on the vestibule [89]. Seven articles addressed the question of the link between the vestibular system and the hypothalamus [2,22,89,90,91,92,93].

We found only seven studies exploring hormonal variations during acute episodes performed by six different research teams [57,60,94,95,96,97]. Very few vestibular pathologies were explored in these studies, since only Menière and BPPV patients were recruited with limited cohorts (21 patients with Menière’s disease and 31 BPPV for the largest number of subjects). Blood samples for parathyroid hormone and vasopressin and saliva samples for cortisol were collected.

To measure normal plasma vasopressin levels, ranging from 0.3 to 3.5 pg/mL, the team of Aoki et al. collected 105 healthy subjects with no history of vestibular or cochlear pathology over a period of the day between 8 and 10 am. This was compared with plasma vasopressin levels collected during the same time period from patients with Ménière’s disease during the vertigo attack, from Ménière’s patients during the remission phase (1 month after the attack), and from patients with other forms of vertigo. Some patients were treated with isosorbide and others with betahistine or difenido hydrochloride. These studies show that vasopressin levels increase significantly during the acute attack in patients with Ménière’s disease, while the change is not significant in patients with Ménière’s disease in remission at any stage of the disease and in patients with other forms of vertigo during the acute attack [57,60].

For their study on Cortisol, the team of Dagilas et al. took samples from 10 volunteers with no particular history, after caloric tests, at the height of their nystagmus. For this purpose, the assays were carried out on plasma taken in the morning between 8 and 9 a.m. Cozma et al. team carried out salivary cortisol measurements in 48 volunteers according to a very precise protocol. Patients with a history of metabolic, cardiovascular, endocrine, vestibulocochlear, or other chronic disease were not included. No participant had taken any anti-inflammatory, antihypertensive, antidepressant, immunosuppressive, or drug treatment in the six months prior to the study. Alcohol, heavy eating, and chocolate were prohibited three days before the samples were taken, while sport and intense emotions were to be avoided only on the day of the tests. Before using the saliva samples, the participants were not allowed to eat, drink coffee, brush their teeth, or do any sport for 30 min beforehand. The samples were taken on the first day, without any vestibular stimulation, at 8 a.m. (at least 2 h after waking up), at 12 p.m., and at 8 p.m., and then the next day 5 min before the caloric challenge, and then at 1, 4, 7, 10, 15, 30, 45, and 60 min afterwards. Both teams found a significant increase in cortisol levels at the time of vertigo compared to the inclusion stage [95,96].

The team of Kahraman et al. sampled 31 patients with benign paroxysmal positional vertigo (BPPV) on the day of their admission after 12 h of fasting, or the following morning for those admitted in the afternoon and at a distance from this event at 6 months [97]. The plasma parathyroid hormone level at this first visit was significantly higher than at the second visit (43.84 pg/mL vs. 25.08 pg/mL) and was significantly increased in 6 patients (>65 pg/mL).

Due to the small number of samples, the results were statistically insignificant and sometimes discordant. Cozma and Dagilas found a significant increase in cortisol levels concomitantly with vertigo attacks [96,97]. Aoki et al. noted an elevated vasopressin level in patients with Menière’s disease and normal levels in patients with BPPV [57,60]. Kahraman et al. found a significantly higher plasma parathormone level compared to patients with BPPV [97].

## 4. Discussion

### 4.1. Why So Little Information?

So why in 2022 is there so little information on the role of hormones in vestibular physiology and pathophysiology? This results from the conjunction of several factors. On the one hand, the low number of research projects devoted to this topic. We carried out a bibliography study over the two last decades in order to evaluate the number of (i) studies related to the dosage of circulating hormones in vestibular disorders patients; (ii) studies on vertigo pharmacology; and (iii) studies related to central compensation and vestibular rehabilitation. The study highlights a very small number of publications devoted to clinical samples of circulating hormones during the vestibular syndrome (Figure 2). While this could be interpreted as a lack of interest on this issue, it could also result from the difficulty of considering it in its entirety and its complexity. Indeed, although there is a clear correlation between certain hormonal profiles (post-menopausal women [34,35,36,37,38,39,40,41,42,43,44], diabetes [51,52,53] and certain vestibular pathologies, such as BPPV or vestibular migraines [54,55]), very few studies have been undertaken to confirm a causal link, using biological, cellular, or molecular investigations in humans or animal models. For example, no study has been undertaken to date to precisely map in animal the expression of hormone receptors along the peripheral and central vestibular networks, or to monitor the vestibular syndrome in animal models of hormonal disorders mimicking human vestibular pathologies. Furthermore, very few clinical studies have been conducted to attempt to establish a direct link between the expression of the vestibular syndrome, in particular acute attacks, and the alteration of the level of circulating hormones. When looking at these studies, it can be seen that different factors that can influence the blood concentration of certain hormones have not always been taken into account. This is, for example, the case for the stage of the hormonal pathology, the age and sex of the patients, the specific diet, the associated pathologies, or the time of the daily dosage. It is therefore crucial, if one wishes to compare the blood level of a hormone such as cortisol, to name but one, to strictly observe the same sampling times. Various factors can lead to biases in the blood determination of circulating hormones, and therefore special precautions must be taken to avoid them.

### 4.2. Hormones and Vestibule: How to Move Forward?

It is obvious that the emergence of specialized research groups worldwide should facilitate a systematic approach to investigate this very large field of hormone–vestibular research. It can be assumed that accumulation of epidemiological studies reinforcing the evidence of a close link between hormonal deregulation and vestibular pathologies will encourage the creation of research structures centered on this specific theme. A multidisciplinary organization of these research groups, gathering scientists with skills in neuroscience, endocrinology, biochemistry, cell and molecular biology, and electrophysiology, and clinicians specialized in the management of the dizzy and unstable patient, will allow taking stock of the R&D needs to meet the strong medical need in the neuro-otology field, and identifying actions on which efforts must be particularly focused. Three of them seem to us to be priorities:

### 4.3. Mapping the Expression of Hormone Receptors over the Whole Vestibular Sensory Network

Until recently, the study of hormone receptors’ membrane expression was primarily carried out using immunocytochemistry or ELISA approaches. One of the main advantages of these methods is that they allow the validation or invalidation of the presence of a sought receptor in a tissue and, in the first case, the precise determination of its location. Among their disadvantages are the dependence on the specificity of the used antibodies and the fact that, in the end, one can only find what they look for. In recent years, the wide diffusion of RNA sequencing approaches has opened a new door to more global research, allowing the identification of all the gene products in a specific tissue, or even in a single cell, with levels of sensitivity and reliability never achieved before. This type of approach therefore today provides the opportunity to discover new hormonal receptors throughout the vestibular network, from the peripheral sensory organs, towards the distal cortical regions involved in the integration and control of the vestibular information. The precise mapping of hormonal receptors will undoubtedly allow us to better anticipate the role of the corresponding hormones, both in the normal function of the vestibular system, but also, of course, in pathological situations.

### 4.4. Specify the Role of Hormones in Vestibular Physiology and Vertigo Pathophysiology

We are currently able, using in vivo and in vitro models, to decipher the mode of action of hormonal mediators within the vestibular sphere. There are some data in the literature concerning the modulation of vestibular function by certain families of hormones. Stress hormones are the most widely documented [98,99]. Among these data, neurohormonal mechanisms normally involved in stress responses were demonstrated to take place after vestibular injury [22]. It was shown that unilateral vestibular neurectomy (UVN) induces long-term activation of the hypothalamic-pituitary-adrenal (HPA) axis with a bilateral increase in the number of neurons expressing the corticotropin releasing factor (CRF), and arginine vasopressin in the paraventricular nuclei of the hypothalamus at 1, 7, and 30 days after UVN. An increase in the synthetic norepinephrine enzyme (dopamine ß hydroxylase: D ß H) in the coeruleus locus was also observed at these same post-injury times. The expression of these neurohormones normalizes when the vestibular syndrome has completely disappeared three months after UVN. It is particularly interesting to note that the compensation for posturo-locomotor deficits follows the same time course. Taken together, these results indicate that activation of the hypothalamic pituitary axis may influence the vestibular plasticity processes underlying behavioral readjustments. In addition, our group has recently demonstrated, for the first time in an animal model of unilateral peripheral vestibulopathy, the benefit of an acute treatment with thyroxine, both for the reduction of the vestibular syndrome and the improvement of balance [3].

### 4.5. Follow the Hormonal Variations over the Different Phases of the Vestibular Syndrome

The interaction between endocrine systems and vestibular syndrome can be considered to be twofold. Vestibular deficits can indeed be the cause of hormonal variations at different times of the vestibular syndrome, such as the acute crisis, or the compensation phase as mentioned above, while specific hormonal profiles could increase the susceptibility of patients to dizziness. This is the case with BPPV, the prevalence of which increases dramatically with age, especially in postmenopausal women. Decreased estrogen and progesterone levels, leading to inner ear microcirculation disorders in postmenopausal women, may explain this prevalence [43,100]. Diabetic patients also seem more likely to develop BPPV. D’Silva et al. reported that BPPV was seen in 46% of patients with type 2 diabetes, versus 37% of people without diabetes [51]. Hyperglycemia is also a risk factor for recurrence of BPPV [51]. BPPV is also associated with thyroid disorders such as goiter, hypothyroidism, thyroiditis, and hyperthyroidism [47]. Ménière’s disease is a common inner ear disease, found in patients who are generally anxious. Stress, by promoting an increase in vasopressin, could be one of the inducing factors at the origin of the onset of Ménière’s disease [59]. Studies have also suggested the association of hypothyroidism with Ménière’s disease [44]. Thus, particular hormonal profiles may promote the expression of different types of vestibular disorders.

In addition, several studies have correlated the plasma expression level of certain neurohormones during the different phases of vestibular pathology. The vestibular syndrome evolves in different phases: it begins with an acute crisis with the appearance of dizziness, most often accompanied by vegetative signs and anxiety. Then comes a phase of functional recovery during which the symptoms gradually regress, thanks to the process of vestibular compensation. The vestibular system has many connections with the hypothalamic–pituitary axis [101], which explains that, at the time of vestibular dysfunction, patients also show changes in their stress hormonal profiles, revealed by changes in the blood concentration of cortisol and ACTH [77]. We also note that the characteristics of central vestibular compensation differ according to the patients, and that they are slower to set up in stressed patients [22].

The actual functional importance of stress neurohormones in vestibular compensation remains unknown. The hypothesis of a deleterious or conversely beneficial action of these neurohormones in the establishment and maintenance of vestibular compensation processes, however, finds some answers in vestibular pathology. After apparently identical vestibular losses, some patients recover completely, while others compensate poorly. Understanding the causes of incomplete or inadequate functional restoration is a major challenge in the management of the dizzy and unstable patients. Stress and its contingent of hormones could be one of the factors behind this differential compensation. A better understanding of the role of the hormonal component in vestibular pathophysiology would be a considerable asset in the development of new antivertigo drugs targeting the endocrine sphere.

## 5. Conclusions

There is a clear scientific need to better understand the interaction between the vestibular system and endocrine system. Along with this scientific need is an urgent medical need to define how the hormonal profile of the vertigo patient is involved in the occurrence of peripheral vestibulopathies, and to what extent hormonal variations accompany the vestibular syndrome expression. Setting up preclinical and clinical studies to answer these questions is essential, on the one hand, to develop new diagnostic tools to discriminate between the different types and stages of vestibular pathologies, and on the other hand, to establish more targeted and more efficient therapeutic approaches to anticipate the onset of a vestibular disorder, to effectively reduce vertigo attacks, and to promote rapid central compensation. We do not know today the proportion of vestibular disorders of hormonal origin, vestibular disorders related to specific hormonal profiles, or those involving hormonal variations. We cannot rule out the idea that these correlations represent a significant proportion of vestibular disorders. It is for this reason, and with the prospect of having a significant impact on the management of dizzying and unstable patients, that a research and development effort must be undertaken in this segment.

## Figures and Tables

**Figure 1 cells-12-00656-f001:**
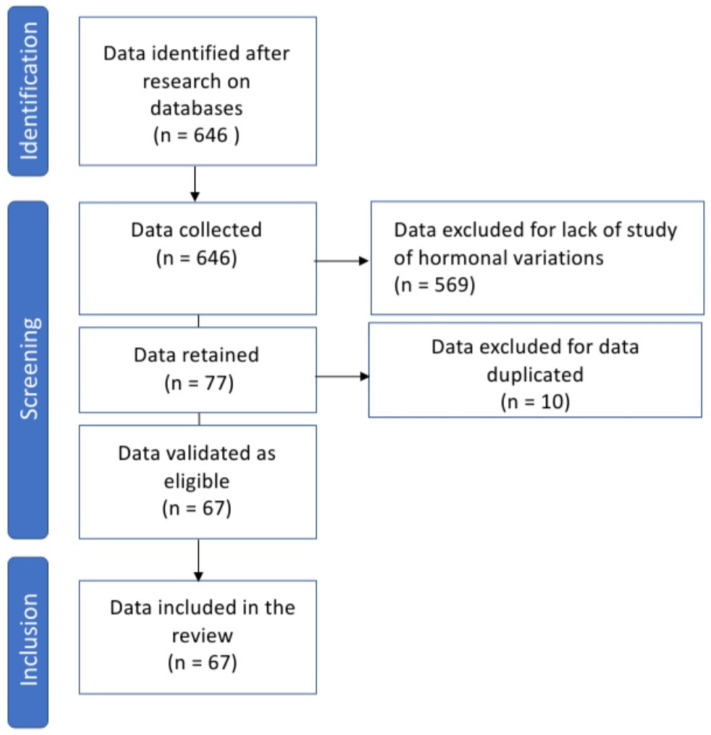
Flowchart presenting the steps of the data collection process (PRISMA method).

**Figure 2 cells-12-00656-f002:**
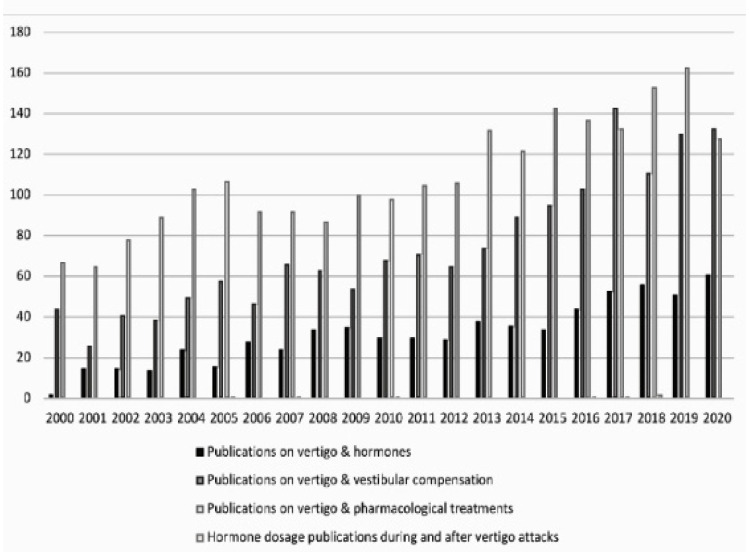
Comparison of the number of publications according to the different searched keywords. Period covered: 2001 to 2021. Horizontal axis: years; vertical axis: number of publications indexed on PubMed.

**Table 1 cells-12-00656-t001:** Table displaying the research method and references of the collected studies associating hormonal variations and vestibular pathologies. Research carried out in February 2022.

Pathology and Method of the Research	Hormonal Variation	References
VPPB		
- BPPV and estrogen, BPPV and peri-menopause, BPPV and postmenopausal,	Estradiol (women in pre and post menopause)	[34,35,36,37,38,39,40,41,42,43,44]
- BPPV and thyroid	Thyroid pathologies	[47,48]
- BPPV and cortisol		
- BPPV and stress hormones		
- BPPV and vasopressin		
- BPPV and insulin	Insulin	[51]
Ménière disease		
- Ménière disease and vasopressin	Vasopressin	[2,56,59,60,61,62,63,64,65,66,67,68]
- Ménière disease and prolactin	Prolactin	[69,70]
- Ménière disease and melatonin	Melatonin	[71]
- Ménière disease and aldosterone	Aldosterone	[72]
- Ménière disease and menopause	Sexual hormones	[73,74,75,76]
- Ménière and stress hormones	Stress hormones	[77,78,79,80]
Vestibular migraine		
- vestibular migraine and hormones	Estradiol	[54,55]
- vestibular migraine and menopause		
- vestibular migraine and estrogen/vestibular migraine and menstrual cycle		
- vestibular migraine and thyroid		
- vestibular migraine and insulin		
- vestibular migraine and cortisol		
Vestibular dysfunction		
- vestibular dysfunction and vasopressin	Vasopressin	[2,12,13,15,81]
- vestibular dysfunction and insulin	Insulin	[52,53]
- vestibular dysfunction and estrogen	Estradiol	[82,83,84,85,86,87,88]
- vestibular dysfunction and menopause		[89]
- vestibular dysfunction and cortisol	Cortisol	
Vestibular system		
Vestibular system and hypothalamus	Hypothalamus	[2,22,89,90,91,92,93]
Acute vertigo		
- acute vertigo and hormones	Different hormones	[57,60,94,95,96,97]
- acute vertigo and cortisol		
- acute and thyroid vertigo		
- acute vertigo and insulin		
- acute vertigo and progesterone		
- acute vertigo and estrogen		
- acute vertigo and vasopressin		

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
