# Peer review of "Vestibular Disorders and Hormonal Dysregulations: State of the Art and Clinical Perspectives"

_cells, 2023, doi:10.3390/cells12040656_

Round 1

Reviewer 1 Report

My  compliments to the authors, this topic is very  diffciult  to  treat . Please see file attached 

Author Response

Vestibular Disorders and Hormonal Dysregulations: State of the Art and Clinical Perspectives The article is very interesting and well written. It is also very accurate and I agree with the suggestions of the authors. I have only very few remarks:

-Lines 171-172 The Hormone and Vertigo team of the GDR2074 CNRS unit (https://gdrvertige.com/en/team/hormones-vertiges/) is one of them. In my opinion , the review does not need the indication of a specific group….It is sufficient to indicate the advantage to create research structures centered on this specific theme. What do you think ?

# We agree with the reviewer comment. We removed from the text indications about this specific group.

-In the same way, Lines 201-202. Among these data, our team has demonstrated, after vestibular injury, neurohormonal mechanisms normally. I would use an impersonal expression and just cite the paper Line 203 . We have shown ….

# Done (now in L2-254)

Reviewer 2 Report

Interesting paper, my suggestion is to revise the References, you mixed articles of different quality and levels of evidence

Author Response

Interesting paper, my suggestion is to revise the References, you mixed articles of different quality and levels of evidence.

# Our intention in this first review paper was not to be exhaustive on the bibliographic references about this specific topic. So we prefer keeping the Bibliography part as it is.

Reviewer 3 Report

This manuscript reviewed the hormonal dysregulations and vestibular disorders, which was well organized and written.

Clinical observations have shown that some vestibular disorders are associated with changes of hormones, but lack of strong evidence. This manuscript reviewed what has been done in this field, and useful information was provided, but more is needed. One suggestion is that the relationship between endocrine and vestibular system should be added in detail.

Author Response

This manuscript reviewed the hormonal dysregulations and vestibular disorders, which was well organized and written. Clinical observations have shown that some vestibular disorders are associated with changes of hormones, but lack of strong evidence. This manuscript reviewed what has been done in this field, and useful information was provided, but more is needed. One suggestion is that the relationship between endocrine and vestibular system should be added in detail.

# To meet the reviewer comment, we had more details on the blood sampling procedures and hormonal dosage in the RESULT part.